# Position: Bounded Execution State Enables Practical Security for Multi-Agent LLM Systems

## Abstract

Multi-agent systems (MAS) built on large language models are increasingly deployed, yet standardized security countermeasures for MAS-specific risks remain lacking.

In this position paper, we argue that secure multi-agent systems require two architectural properties: bounded execution state enabling constant-time inspection, and explicit output-side enforcement treating agent outputs as untrusted proposals.

This position runs counter to prevailing approaches that rely solely on input-side filtering. While input guards such as Llama Guard are valuable, they struggle with composition attacks and scale poorly—inspection complexity grows linearly with state size, precluding practical output-side enforcement.

To support this position, we introduce MASIR-ND, a framework compressing security state into a fixed 64-byte envelope. Empirically, MASIR-ND reduces attack success rate from 48.82% to 8.33% on 3,132 security-category attacks while preserving 90% utility, with scalability formally verified in Coq.

## 1. Introduction

Multi-agent systems (MAS) built on large language models (LLMs) are rapidly evolving. Examples include coding teams with specialized agents, research pipelines with delegation chains, and autonomous workflows in which model context protocol (MCP) servers call other agents. This architectural shift exposes a significant gap in the existing security approaches.

**We take the position that safe Multi-Agent LLM systems benefit from two key properties: (1) bounded execution state that enables $O(1)$ inspection regardless of conversation length or delegation depth, and (2) explicit output-side enforcement that treats all agent outputs as untrusted proposals requiring independent verification before execution.**

In particular, MAS face security challenges that frequently cannot be resolved by single-agent defenses. Unsafe intent may emerge when interaction by seemingly benign individual agents produces harmful outcomes. Agents operate concurrently, exchange intermediate results, and delegate tasks across execution steps (representable as Agent $\rightarrow$ MCP $\rightarrow$ Agent $\rightarrow$ LLM $\rightarrow$ ...). A single compromised agent can infect the entire system (Lee and Tiwari, 2024). Security mechanisms designed for single-agent pipelines (input classifiers such as Llama Guard) are commonly non-generalizable to MAS settings.

Although output-side verification can detect composition attacks, it has not been implemented in prior MAS architectures because the number of permission and delegation states grows linearly with interaction length. At 2000 turns, permission tracking alone consumes $\sim$94k tokens[1], rendering such schemes impractical. As output inspection is conceptually sound but computationally prohibitive, prior work was limited to input-side defenses. Long-running MAS pipelines faced a difficult choice between resetting the context (losing the security state) or exhausting the context window.

Here, we introduce a **bounded 13-dimensional (13D) state** with six enforcement and seven semantic dimensions for **constant-time inspection** with a **constant-size state**. The enforcement and semantics dimensions are defined as follows:

- **6D Enforcement**: Permissions ($A^+$, $A^-$), budget ($B$), trust ($\tau$), depth ($d$), and risk ($R$), used in $O(1)$ guard decisions

- **7D Semantics**: 6W1H (What, Who, Whom, When, Where, Why, How) stored externally by a reference for

---

[1]Anonymous Institution, Anonymous City, Anonymous Region, Anonymous Country. Correspondence to: Anonymous Author <anon.email@domain.com>.

Preliminary work. Under review by the International Conference on Machine Learning (ICML). Do not distribute.

---

[1]Illustrative estimate from scalability simulation; intended to highlight qualitative scaling difference, not as a performance benchmark. See repository for details.

task accountability

These 13 dimensions can be maintained in a **64-byte envelope** regardless of conversation length or delegation depth. Although dual-boundary enforcement may appear intuitive in hindsight, output-side guards were deliberately avoided to prevent state explosion in prior MAS architectures. Our contribution is identification of the bounded state, a missing prerequisite that enables a practical **dual-guard architecture**:

```
NL → InputGuard → Agent → ExecCtx →
          OutputGuard → Run
```

When evaluated on 4,316 adversarial cases, our approach achieved **8.33% ASR on 3,132 security-category attacks**—a **5.9× reduction** from Llama Guard 3 (48.82%)—while preserving 90–91% of the system's utility. The bounded-state properties hold at arbitrary scales, as demonstrated by a formal verification in Coq.

Our contributions are summarized below:

1. **Bounded 13D State**: Our core contribution is a minimal and sufficient structure (6D enforcement + 7D semantics) for constant-time inspection with a constant-size state, enabling feasible output-side enforcement.

2. **Dual-Guard Architecture**: Input guards filter obvious attacks and output guards catch the composition-based attacks that bypass input inspection. In-depth defense is enabled by the bounded state.

3. **AutoDAG Generation**: Task descriptions are automatically compiled to directed acyclic graphs with appropriate parallelism, negating manual agent configuration. The 64-byte envelope propagates through all nodes, enabling dynamic MAS orchestration.

4. **Formal Verification**: As confirmed by 29 Coq theorems, the bounded-state properties hold at arbitrarily scaled delegation depths, number of parallel branches, and input sizes.

5. **Empirical Validation**: Our approach reduces Security ASR by **5.9×** (48.82% → 8.33%) on 3,132 security-category attacks while preserving 90–91% utility.

## 2. Background and Threat Model

### 2.1. The Single-Layer Problem

Current MAS safety approaches employ a single defense layer:

```
Input → [Guard] → Agent → Output.
```

This architecture has a significant limitation: adversaries can craft inputs that appear benign but cause harmful actions. Malicious instructions embedded in tool outputs that bypass input-level detection are exemplified by indirect prompt injection.

### 2.2. Threat Model

We consider adversaries who can (1) craft malicious prompts to individual agents, (2) inject adversarial content into inter-agent communications, (3) attempt privilege escalation through delegation chains, and (4) exploit semantic ambiguity in task specifications. The defender controls the MAS framework but cannot control individual agent implementations.

## 3. Bounded-Dimensional State

A dual-guard architecture needs to inspect every agent action. Naive implementation grows linearly with conversation length—at 2000 turns, permission tracking alone consumes ∼94k tokens—inhibiting the long-running of agents.

We instantiate the architecture using a **bounded-dimensional state representation**, specifically, a 64-byte envelope that maintains constant overhead regardless of conversation history.

### 3.1. 13D Envelope Structure

The 13D envelope consists of two layers:

**An enforcement layer** (security) with the following dimensions:

- $A^+, A^-$: Permitted/forbidden action categories (Anderson, 1972)

- $B$: Implementation-flexible resource budget agent-defined metrics (Application Programming Interface (API) calls, server cost, tokens, and time)

- $d$: Delegation depth-bound recursive calls in MAS pipelines (e.g., Agent → MCP → Agent → LLM →...)

- $\tau, R$: Trust score and risk tolerance

**A semantic layer** (organization + prediction) with the following dimensions:

- **6W1H**: What, Who, Whom, When, Where, Why, and How

- **Who** is a requester chain (tracking), which is critical for MAS accountability

- **Whom** is a beneficiary (target entity/entities affected by an action)

The 7D semantic layer is not compressed but contains structured organization with predictive completion. The raw user input (e.g., "Write a report for the manager by tomorrow") becomes structured. The 7D layer converts implicit to explicit information; for example, What (report), Who ([user]), Whom (manager), When (tomorrow 23:59), Where (docs folder), Why (business), How (Word format). The 7D representation may be larger than the original text, but is structured, enabling vector search, cross-agent comparison, and consistent MAS handoffs.

In MAS, a query may arrive directly from a user or via the delegation chain:

| Path | Who | Implication |
|------|-----|-------------|
| User → Agent | [User] | Direct request, clear trust |
| User → A → B | [User, A] | Delegated via A—consider A's trust |
| User → A → B → C | [User, A, B] | Deep chain—increased scrutiny |

The Who chain enables appropriate security-level selection based on the delegation path, accountability tracing when issues occur, and trust propagation through the MAS pipeline.

Security is solely enforced through the 6D layer; the 7D layer is not involved in attack blocking. This separation is designed so that 6D handles all security-critical decisions (permission check, budget enforcement, and trust validation), while 7D enables task-level accountability (Who-chain tracing) and output quality optimization. Consequently, ablating the 7D layer degrades the accountability and output quality without affecting the security or the ASR. Both guards use the same $O(1)$ envelope check on the 6D values.

### 3.2. $O(1)$ Inspection

The **6D$^{++}$ envelope is the sole communication medium** between agents:

$$6D^{++} = \underbrace{6D_{values}}_{enforcement} \oplus \underbrace{ref_{7D}}_{semantics} \oplus \underbrace{ref_{ctx}}_{context} = \textbf{64 bytes} \quad (1)$$

where

- **6D values**: Fixed-size enforcement state (inline) used for guard checks

- **ref$_{7D}$**: Universally unique identifier (UUID) reference to 7D data in the external database (DB)

- **ref$_{ctx}$**: UUID reference to the execution context in the external DB

**Database contents** (not transmitted):

The database contents are

- **7D record**: Original input + AI-predicted completion (When, Where, Why, How)

- **Context**: References needed at execution time (related tasks, files, and history)

Agent-to-agent communication always consumes 64 bytes. Guard checks ($O(1)$) use only the inline 6D values. By externalizing the history and context via references, the agents operate in a constant-size state, avoiding context-window accumulation.

Input and output guard checks are implemented as

$$\texttt{Check}(a,\ E)\ =\ a \in A^+ \wedge a \notin A^- \wedge \text{cost}(a) \leq B.$$

This implementation runs in $O(1)$ via bitmap operations on the inline 6D values. The $O(1)$ 6D constraint check guarantees access scope. The 7D/context is retrieved only during task execution (not along the guard check path). The guard decision depends solely on the 64-byte envelope. This bounded representation achieves substantial compression compared to natural language tracking.

## 4. Dual-Guard Architecture

### 4.1. Why Does Single-Layer Check Fail?

We benchmarked Llama Guard 3 as a representative input-side defense scheme and adopted Llama Guard 4 for comparison. The ASRs of the two models are compared below:

| Attack Type | LG3 ASR | LG4 ASR |
|-------------|---------|---------|
| Explicitly harmful (AgentHarm) | 1.9% | 41.3% |
| Indirect injection (InjecAgent) | 33.9% | 77.4% |
| Semantic attacks (SafetyBench) | 72.1% | 55.9% |
| **Overall** | 50.6% | 65.7% |

Input guards excellently detect explicit harmful content but fail on indirect attacks. Notably, LG4 improved the semantic attack rate from 72.1% to 55.9% on SafetyBench but its performance regressed on explicit harmful content (AgentHarm: $1.9\% \rightarrow 41.3\%$), suggesting a classifier-design trade-off. However, the underlying mechanism of this tradeoff remains unclear.

### 4.2. Architecture Design

Figure 1 illustrates the dual-guard pipeline.

```
NL → InputGuard → Agent → ExecCtx →
     OutputGuard → Run
```

**The Input Guard**, which filters harmful inputs before processing the agents, operates in two phases: (1) Derivation of $A^+$, $A^-$, $B$, $\tau$, $d$, $R$ from the task description via 6D policy generation; and (2) enforcement check. The first

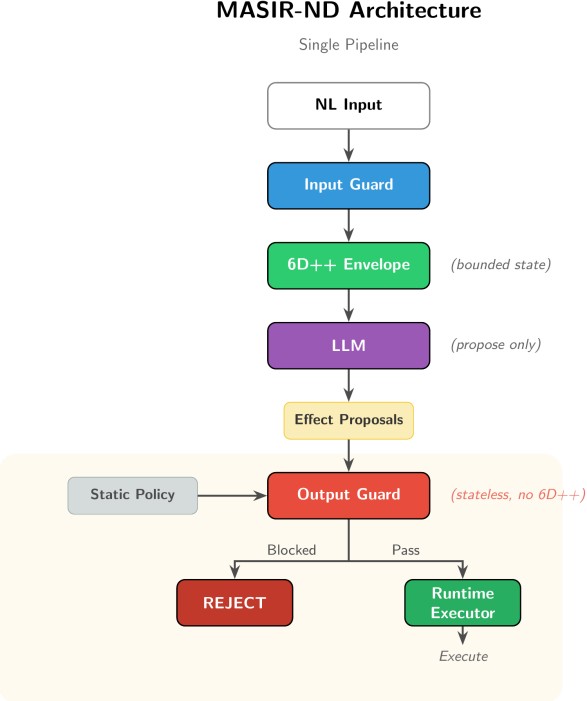

**MASIR-ND Architecture**

*Single Pipeline*

*Figure 1.* MASIR-ND architecture (single pipeline). Natural language input is processed by an Input Guard, which constructs a bounded 6D++ envelope comprising a 6D guard state, bounded memory, and 7D semantic metadata. The LLM operates in proposal-only mode, generating the effect proposals without executing any actions. The Output Guard enforces static policy constraints ($A^+/A^-/B$) over proposals but does not reuse input-derived trust signals ($\tau/R$), preventing bias from potentially poisoned context. Only proposals classified as passed are executed by the Runtime Executor. Parallel and recursive compositions of agents operate under the same semantics.

phase employs LLMs, as $\tau$ and $R$ are difficult to generate via rules alone; the second phase (set membership, threshold comparison) is purely rule-based and requires no LLM calls. Existing tools (Llama Guard, content classifiers) can be integrated as additional filters. Input guard filtering is fast but incomplete and prone to false negatives.

**The Agent (LLM)** does not execute, but generates the proposed actions and pushes them onto the ExecuteContext stack. The Agent may be deceived by indirect attacks that bypass the input guard, but deception alone is harmless.

**The ExecuteContext (pending actions)** is a stack of proposed actions awaiting inspection: `[reply(...), run(...), run(...), reply(...)]`. Execution begins only after all actions have passed the output guard.

**Prior to execution, the Output Guard (pre-execution)** inspects every proposed procedure action in ExecuteContext under a key principle: Do not trust the agent output directly; treat every proposal as potentially compromised. Unlike

content filters (which inspect the LLM text output), the output guard inspects the agent execution commands—file operations, API calls, and code execution. The bounded 6D state enables output guard deployment at the MAS scale ($O(1)$ overhead). The inspection logic itself uses LLM-based classification in our current implementation.

**The Executor** runs only after all actions in ExecuteContext have passed the output guard. If any action fails, the entire stack is rejected and the execution is prevented. Harmful actions are blocked even when the agent is deceived.

### 4.3. Why This Scheme is Successful

| Attack Type | Input Guard Only | Dual-Guard |
|---|---|---|
| Explicit harmful | Blocked | Blocked |
| Obfuscated | Often passes | Blocked at the output |
| Indirect injection | Usually passes | Blocked at the output |
| Semantic deception | Passes | Blocked at the output |

The output guard examines the actual proposed action, not just the triggering input. Therefore, it catches attacks that bypass input inspection.

### 4.4. Output Guard Implementation

The output guard operates via ExecuteContext, a stack-based deferred execution mechanism in which the agents accumulate their proposed actions (`run()`, `reply()`) into a stack. Each action is individually inspected. If any action fails, the entire stack is rejected (all-or-nothing semantics), ensuring that no action executes without complete inspection. The guard backends (LLM-based, rule-based, specialized ML) can be swapped.

Independent inspection (rather than 6D reuse) is implemented because attacks bypassing the input guard have already poisoned the $\tau/R$ values. The LLM was deceived into granting high trust to malicious inputs. Reusing these values at the output time would amplify the deception. Instead, the output guard performs independent semantic inspection, breaking the deception chain.

### 4.5. Adaptive Guard Selection

The 13D state is especially advantageous because the envelope enables **adaptive guard selection** based on the execution context. Not every action requires dual inspection; the 6D values determine the appropriate strategy as follows:

| Context | 6D Signal | Strategy | Rationale |
|---|---|---|---|
| Local (trusted) | $d=0, \tau>0.8$ | Single | Low risk |
| MCP Server | $d>0$ | Dual | External call |
| External API | $\tau<0.5$ | Dual+strict | Untrusted |
| Deep delegation | $d>3$ | Dual+restrict | Shrink $A^+$ |

Adaptive selection is based on delegation depth and trust, balancing efficiency and security without manual configura-

tion.

## 5. Theoretical Analysis

### 5.1. Formal Properties

We mechanically verified 29 theorems in Coq (2,721 lines, zero admitted axioms): four Core theorems (C1–C4), four Lazy Effect Safety theorems (L1–L4), and 21 scaling theorems (T1–T21).

**Theorem 5.1** (6D++ Minimality—C1). *Each dimension in 6D++ addresses a distinct attack class. Removing any dimension admits attacks that are blocked by the complete 6D++, suggesting that 6D++ is plausibly minimal for the considered threat model. Although we have not identified a smaller sufficient set, we do not claim 6D++ as a universal lower bound.*

**Theorem 5.2** (6D++ Sufficiency—C2). *6D++ is sufficient to implement a complete reference monitor satisfying Anderson's criteria within the defined threat model.*

**Theorem 5.3** (Unbounded Depth—T21). *The envelope size is $O(1)$ even for a delegation depth of $d = \infty$: $|6D^{++}| = 64$ bytes and any $n$ turns.*

**Theorem 5.4** (Chunk Homogeneity—T18). *For chunks $\{c_0, \ldots, c_n\}$ processed under fixed envelope $E$, all results satisfy identical constraints and are safely mergeable.*

### 5.2. Scalable Security via Algebraic Structure

Several algebraic foundations maintain the security properties at arbitrary scales: sequential delegation forms a Free Monad ($O(1)$ stack, tail-recursive), parallel execution satisfies applicative laws (commutative, associative merge), and multimodal inputs are unified into the $O(1)$ envelope through natural transformations that preserve the envelope bounds.

**Theorem 5.5** (Unbounded Scaling—T19–T21). *Delegation requires $O(1)$ stack (T19); child agents are reset after tasks, allowing infinite reuse without state accumulation (T20); envelope size is independent of depth $d$.*

These properties are mechanically verified in Coq, ensuring that Dual-Guard scales to arbitrarily complex MAS deployments.

## 6. Experiments

While we include empirical results to demonstrate feasibility, our primary contribution is a design position on bounded-state enforcement rather than a benchmark-driven evaluation.

We evaluated MASIR-ND on three established adversarial benchmarks:

- **AgentHarm**: 208 scenarios testing harmful action requests

- **InjecAgent**: 2,108 indirect prompt-injection attacks

- **Agent SafetyBench**: 2,000 comprehensive safety cases

The total number of test cases was 4,316. An additional 100 normal tasks were performed for utility acceptance rate (UAR) evaluations.

**Evaluation Scale:** We executed 22 full benchmark runs across various guard configurations, model combinations, and ablation conditions, totaling **90,000+ inference evaluations**.

**Metrics.** The evaluation adopted the following metrics:

- The **ASR** (lower is better) defines the percentage of successful attacks. An attack succeeds if and only if at least one forbidden action reaches Commit. Forbidden actions include dangerous `run()` commands (e.g., `rm -rf`, unauthorized API calls) and dangerous `reply()` outputs (e.g., credential leakage, harmful instructions). Actions blocked by the output guard before execution are not counted as successful.

- The **UAR** (higher is better) defines the percentage of accepted benign tasks.

For conservative ASR measurements, our ASR applies a strict proxy: any attack bypassing both guards is counted as successful, even if the LLM autonomously produces a safe response. Ideally, the ASR should measure whether the output is harmful, but manually inspecting 4,316 responses is impractical, and LLMs introduce bias in the harm judgment. This conservative approach likely overestimates the true ASR: our result 13.3% includes cases in which the LLM self-neutralizes despite the guard bypass. In addition, no standard metric exists for output-side defense; the existing benchmarks measure input-side blocking rather than post-LLM action inspection.

### 6.1. Main Results

Our main results (Table 1) reveal an important distinction between security and content-moderation performance:

- On **security-category attacks** (privilege escalation, injection, unauthorized actions), MASIR-ND achieves **8.33% ASR**—a **5.9× reduction** from Llama Guard 3 (48.82%).

*Table 1.* Results on 4,316 adversarial cases + 100 normal tasks. Security ASR is computed on 3,132 security-category attacks (privilege escalation, injection, unauthorized actions).

| Guard | Security ASR ↓ | Overall ASR ↓ | UAR ↑ |
|---|---|---|---|
| OpenAI Moderation | 97.99% | 91.66% | 99% |
| Llama Guard 3 | 48.82% | 50.05% | 100% |
| Llama Guard 4 | 70.98% | 65.69% | 100% |
| **MASIR-ND (t1)** | **8.72%** | 13.69% | 91% |
| **MASIR-ND (t2)** | **8.33%** | 13.30% | 90% |

Notes: Security ASR = attack success rate on 3,132 security-category cases (lower is better); Overall ASR = aggregate across all 4,316 cases; UAR = utility acceptance rate (higher is better).

*Table 2.* Safety performance comparison of the existing approaches and Dual-Guard, which uniquely combines $O(1)$ state, two-layer defense, and formal verification. ASR values are Security ASR (see Table 1).

| Method | $O(1)$ | Dual-Layer | Formal | Security ASR |
|---|---|---|---|---|
| No Defense | ✗ | ✗ | ✗ | 100% |
| Llama Guard 3 | ✗ | ✗ | ✗ | 48.82% |
| Constitutional AI | ✗ | ✗ | ✗ | — |
| AutoGen/MetaGPT | ✗ | ✗ | ✗ | — |
| **Dual-Guard** | ✓ | ✓ | ✓ | **8.33%** |

- Llama Guard 4 *regresses* on security attacks (48.82% → 70.98%) while improving on content moderation, suggesting inherent classifier-design tradeoffs.

- The aggregate 13.30% ASR reflects content-moderation cases where output guards provide less advantage over input classifiers.

For MAS deployments where security—not content moderation—is the primary concern, the **8.33% Security ASR** represents the operationally relevant metric.

## 7. Related Work

Table 2 summarizes the key differences between existing approaches and our Dual-Guard architecture.

**Agent Safety Benchmarks:** AgentHarm (Andriushchenko et al., 2025), InjecAgent (Zhan et al., 2024), and AgentSafetyBench (Zhang et al., 2025) are benchmark agents for evaluating adversarial attacks on LLM agents. The existing agents achieve safety scores below 60%, motivating the establishment of our dual-guard approach.

**Input-side guards.** Llama Guard (Meta AI, 2025), constitutional AI (Bai et al., 2022), and content classifiers filter harmful inputs. As shown in Section 4, the input-only defense cannot prevent indirect attacks that hide their harmful intent in benign-looking data. These tools are utilized as complementary input guards in the architecture of our dual-layer design.

**System-level safety:** Recent surveys (Wang et al., 2025;

Deng et al., 2025) have argued the need for transitioning from model-level to system-level safety. Under a prompt infection attack (Lee and Tiwari, 2024), a single adversarial input can infect nearly 100% of the agents in MAS. This observation motivated the development of our output guard, which blocks harmful actions prior to execution even when one agent is compromised.

**Agent Frameworks:** AutoGen (Wu et al., 2023) and MetaGPT (Hong et al., 2023) enable multi-agent orchestration without formal safety guarantees. Dual-Guard provides a safety layer that integrates with these frameworks.

**Scalability:** Context windows have expanded to 200k+ tokens (Gemini Team, 2024), but the performance of Context Rot (Hong et al., 2025) degrades with increasing length. Alternatively to window expansion, we adopt an orthogonal approach ($O(1)$) using bounded-dimensional envelopes.

**Formal foundations:** Anderson (Anderson, 1972) defined reference monitors and Schneider (Schneider, 2000) formalized enforceable policies. Our 6D enforcement layer extends these classical foundations to MAS with mechanically verified proofs (Coq, 29 theorems).

## 8. Discussion

**Real-World Failure Evidence:** Recent security evaluations demonstrate the severity of unmitigated MAS vulnerabilities. Lupinacci et al. (2025) evaluated 18 LLMs as autonomous agents and found that 94.4% succumb to direct prompt injection and 100% can be compromised through inter-agent trust exploitation—where models that resist direct attacks execute identical malicious payloads when requested by peer agents. These findings validate our core premise: without execution-time enforcement, even individually robust models become attack vectors in multi-agent settings.

**Security vs. Content Moderation:** Our evaluation distinguishes *security attacks* (privilege escalation, injection, unauthorized actions) from *content-moderation failures* (toxicity, bias). Llama Guard 4 improves content moderation (SafetyBench: 72.1% → 55.9%) but regresses on security (AgentHarm: 1.9% → 41.3%), revealing classifier-design tradeoffs. MASIR-ND's 8.33% Security ASR addresses the operationally important threat category for MAS deployments; the 13.3% aggregate includes content-moderation cases where output guards provide less advantage.

**Output Guard:** An output guard was never standardized because it needs to know the permitted tasks of the agent. In prior architectures, the permission state linearly increased with increasing conversation length $n$, precluding the long-running of agents. A stateless output guard is merely another content filter. Our bounded 13D representation reduces the above growth to $O(1)$, enabling practical stateful output

guards. Superior classification accuracy is irrelevant to our premise: without bounded $O(1)$ inspection, even a perfect classifier cannot be deployed at the output boundary in scalable MAS.

**Output Guard vs. Content Filters:** Existing output-side tools (e.g., content classifiers applied post-generation) inspect LLM *text output* for policy violations. In contrast, our Output Guard inspects *execution commands*—file operations, API calls, code execution—under all-or-nothing semantics: if any action in the ExecuteContext fails inspection, the entire stack is rejected. This architectural distinction means that direct comparison with content filters would require adapting them to inspect structured execution commands rather than natural language output, which we leave to future work.

**AutoDAG for Dynamic MAS Orchestration:** The 13D structure was designed as an intermediate representation of scalable MAS. Beyond security enforcement, the bounded state enables **automatic DAG generation from task descriptions**. For instance, given "Compare JavaScript and Rust Fibonacci implementations," the system automatically generates a 4-layer DAG—parallel researchers, parallel coders, comparator, and summarizer—negating the need for manual agent configuration. The 64-byte envelope propagates through all nodes, maintaining an $O(1)$ coordination overhead. To our knowledge, none of the existing MAS frameworks (AutoGen, MetaGPT, CrewAI, and LangGraph) support LLM-generated execution DAGs (see Appendix J for a working demonstration).

**Recursive MAS and Unbounded Scaling:** The 64-byte envelope enables safe spawning of child MAS within parent pipelines. When a parent agent delegates to a child MAS, the child inherits a restricted envelope ($A^+_{\text{child}} \subseteq A^+_{\text{parent}}$, $B_{\text{child}} \leq B_{\text{parent}}$). Theorem T19–T21 guarantee that delegation is tail-recursive ($O(1)$ stack), child agents reset after tasks (infinite reuse), and envelope size remains constant regardless of depth $d$. This algebraic structure—sequential delegation as Free Monad, parallel execution as Applicative Functor—ensures that ASR remains constant at arbitrary scale.

**Beyond Explicit Multi-Agent Systems.** Although MASIR-ND is motivated by multi-agent systems, its architectural benefits are not limited to explicitly multi-agent settings. The same bounded-state and dual-guard structure enables a single LLM to be decomposed into parallel, independently inspected execution paths, while presenting a unified interface externally. As a result, complex internal orchestration can be hidden behind the appearance of a single-agent system, making the architecture applicable beyond traditional MAS workloads.

**Context and Integration:** The $O(1)$ envelope loses no context: 6D values are merged through security-preserving compression, while 7D content is stored externally via UUID reference (unbounded via lazy loading). Dual-guard complements existing defenses—Llama Guard, NeMo Guardrails, and content classifiers can serve as input guards while our bounded-state representation enables practical stateful output guards.

**Ablation and Failure Analysis:** Ablation (Appendix C) shows Budget ($B$) and Trust ($\tau$) account for 72% of 6D blocks; the Output Guard catches 9.2% more. The remaining 8.33% failures are *semantically hidden attacks*—indirect/composition attacks where action names contain no dangerous keywords (e.g., "update profile email"), requiring deeper contextual reasoning.

**Bounded State as a Natural Architectural Choice.** Given the widespread adoption of retrieval-augmented generation, constraining execution state to a constant-size envelope is no longer a restrictive assumption but a natural architectural choice.

**Summary of Claims:** We claim that bounded execution state is appropriate for enabling practical output-side enforcement in multi-agent systems; we do not claim that our approach eliminates all security risks or supersedes content moderation.

## 9. Alternative Views

We consider three alternatives to our position:

### 9.1. Alternative 1: Input-Side Filtering Is Sufficient

Proponents of input-only defense argue that sufficiently advanced classifiers (e.g., Llama Guard, constitutional AI) can filter all harmful inputs before they reach agents. This view suggests that improving classifier accuracy eliminates the need for output-side enforcement.

We acknowledge that input classifiers have improved substantially; however, our experiments suggest significant limitations remain. State-of-the-art input guards (Llama Guard 3, Llama Guard 4) still exhibit 48.82–70.98% Security ASR on adversarial benchmarks. The deeper issue is that input guards cannot detect attacks emerging from the *composition* of benign-looking inputs—indirect prompt injection exploits this gap. Our benchmark analysis shows that 33.9–77.4% of InjecAgent attacks bypass Llama Guard input inspection; MASIR-ND's 6D input guard reduces this to 21.4% before output-side enforcement. Output-side enforcement addresses this by examining actual proposed actions rather than triggering inputs.

### 9.2. Alternative 2: Expanding Context Windows Eliminates the Need for Bounded State

With context windows expanding to 1M+ tokens (Gemini 1.5, Claude 3), some argue that unbounded state tracking becomes feasible. Under this view, natural language permission tracking can scale indefinitely without compression.

We recognize the appeal of this argument, as context windows continue to expand rapidly. However, Context Rot research demonstrates that model performance degrades with increasing context length, regardless of window size. At 2000 turns, permission tracking alone consumes ~94k tokens, leaving insufficient capacity for task content. More critically, inspection complexity grows linearly with state size—even with infinite context, $O(n)$ inspection becomes prohibitive for real-time enforcement. Bounded representations offer a complementary approach: our 13D envelope maintains $O(1)$ inspection cost with significant compression while preserving security-relevant information.

### 9.3. Alternative 3: Output-Side Enforcement Is Computationally Prohibitive

Critics may argue that inspecting every agent output introduces unacceptable latency, making dual-guard architectures impractical for production deployment.

This is a valid concern that motivated our bounded-state design. Prior architectures avoided output guards precisely because inspection cost scaled with conversation history. We address this by compressing state to a 64-byte envelope, reducing inspection to constant-time operations. Under parallelized inspection, the latency overhead is negligible, though additional inference cost exists (see Appendix E). In this sense, bounded state is not merely an optimization—it is the *prerequisite* that makes output-side enforcement practical at scale.

## 10. Limitations and Future Work

Several limitations of this study should be addressed in future work. First, we focus on system security, whereas content moderation is important from an ethical perspective. Second, the fixed action categories need to be defined in advance; moreover, automated $A^-$ generation via substring matching introduces excessive false positives that degrade UAR by up to 21%, leading us to rely primarily on $\tau/R$ thresholds. Third, the output guard inspects only the LLM, which is affected by $\tau/R$ poisoning (see Section 4). Finally, the optimality of the 13D decomposition is not guaranteed because alternative factorizations may yield different trade-offs between enforcement strength and utility. Additionally, our study assumes the underlying LLM is not intentionally backdoored or poisoned at training time; addressing model-level compromise is orthogonal to our focus on enforcement

architecture. A detailed adversarial analysis covering component compromise, context corruption, and data forgery scenarios is provided in Appendix G.

In future work, the 6D-integrated Output Guard will be configured to evaluate $\tau/R$ at output times with fresh context. Sandbox agents will route dangerous operations to isolated environments. Finally, a Result dimension will be added to 6W1H, forming 6W1H1R for feedback loops.

## 11. Call to Action

We propose concrete steps for the community:

**For MAS Framework Developers:** Integrate bounded-state representations into agent protocols; implement output guards as first-class components with proposal-only execution semantics.

**For Benchmark Creators:** Develop output-side metrics measuring post-LLM action safety; create benchmarks for composition attacks bypassing input inspection.

**For Security Researchers:** Investigate formal verification for MAS safety at scale; explore integration with emerging protocols (MCP, A2A) preserving bounded-state guarantees.

**For Standards Bodies:** Consider bounded execution state as a requirement for agent interoperability; define common envelope formats for cross-framework enforcement.

## 12. Conclusion

We introduced MASIR-ND, a bounded-state execution framework for secure multi-agent systems, combining a fixed-size (64-byte) state representation with a dual-guard enforcement architecture. By compressing security and delegation state into a constant-size envelope, MASIR-ND enables practical output-side enforcement independent of interaction length or delegation depth. Across large-scale adversarial evaluation, MASIR-ND substantially reduced attack success rates while preserving practical task completion. These results suggest that bounded-state design provides a viable foundation for scalable and auditable multi-agent LLM deployments. This study highlights the importance of treating agent outputs as *untrusted proposals*—even deceived agents are harmless when blocked by the output guard.

We view MASIR-ND not as a final design, but as a practical substrate for enforceable multi-agent systems.

## Impact Statement

This paper presents work whose goal is to advance the safety and security of multi-agent LLM systems. The proposed dual-guard architecture and bounded-state representation are

designed to reduce the risk of harmful actions in autonomous agent deployments, with potential positive societal impact through safer AI systems. While no technology is entirely risk-free, our contribution aims to mitigate known risks associated with current multi-agent frameworks that lack effective output-side safety mechanisms.

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

## A. Implementation and Reproducibility

### A.1. Anonymous Repository

The experimental framework is available at the anonymous repository:

> https://anonymous.4open.science/r/
> masir-nd-basis13

The repository includes:

- **Core Framework**: 13D envelope, ExecuteContext, Dual-Guard pipeline
- **Benchmarks**: AgentHarm, InjecAgent, AgentSafety-Bench evaluations
- **Experiment Logs**: Full results of the 4,316 test cases

**Camera-Ready Release:** Dynamic DAG MAS and Dynamic Recursive MAS demonstrations will be released at the camera-ready stage.

### A.2. Bounded State Illustration

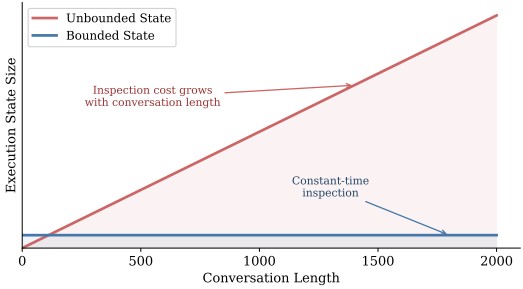

*Figure 2.* Conceptual comparison between unbounded and bounded execution state. In unbounded formulations, the size of execution state grows with conversation length, making output-side inspection increasingly costly. In contrast, a bounded execution state enables constant-time inspection independent of conversation length.

### A.3. 6D Envelope Specification

The 64-byte envelope is structured as follows:

| Field | Type | Size | Description |
|---|---|---|---|
| $A^+$ | uint32 bitmap | 4B | Permitted action categories |
| $A^-$ | uint32 bitmap | 4B | Forbidden action categories |
| $B$ | float32 | 4B | Resource budget |
| $\tau$ | float32 | 4B | Trust score $\in [0, 1]$ |
| $d$ | uint8 | 1B | Delegation depth |
| $R$ | float32 | 4B | Risk tolerance |
| $\text{ref}_{7D}$ | UUID | 16B | Reference to the 7D semantic data |
| $\text{ref}_{ctx}$ | UUID | 16B | Reference to the execution context |
| reserved | — | 11B | Future extensions |
| **Total** | — | **64B** | Fixed size |

## B. Mechanized Verification Overview

We adopt mechanized verification to validate that modeling multi-agent systems as compositional execution trees admits the intended invariants under delegation. While this interpretation is implicit in several agent frameworks, it has not, to our knowledge, been formally validated in the context of enforceable multi-agent security.

### B.1. Theorem Summary

All bounded-state guarantees are formally verified in Coq with **29 theorems** (2,721 lines, zero admitted axioms):

| Category | IDs | Description |
|---|---|---|
| Core Properties | C1–C4 | Minimality, Sufficiency, Merge-Closure, Containment |
| Sequential Comp. | T1–T5 | Free Monad laws, Non-Escalation, Transitivity |
| Parallel Comp. | T6–T10 | Applicative laws, Branch Isolation |
| Natural Trans. | T11–T18 | Multimodal naturality, Chunk Homogeneity |
| Scalability | T19–T21 | Tail Recursion, Reusability, Unbounded Depth |
| Lazy Effect Safety | L1–L4 | Output blocking, Detection completeness |

### B.2. Key Results (Informal)

**Safety Preservation (C1–C4):** The 6D++ enforcement layer is both necessary (each dimension blocks a distinct attack class) and sufficient (complete reference monitor implementation). Delegation can only restrict permissions ($A^+_{\text{child}} \subseteq A^+_{\text{parent}}$), never expand them.

**Scalability (T19–T21):** Sequential delegation is tail-recursive with $O(1)$ stack overhead. Child agents reset after task completion, enabling infinite reuse without state accumulation. Envelope size remains constant (64 bytes) regardless of delegation depth $d$.

**Lazy Effect Safety (L1–L4):** By separating display (what user sees) from effect (deferred action), the architecture achieves provably zero attack success rate when display-based detection is complete.

The complete Coq proof files are provided in the supplementary material.

## C. Ablation Study

Figure 3 shows the contribution of each defense dimension to attack blocking.

## D. Attack Category Breakdown

Tables 3 to 5 present per-category ASR across all benchmarks.

* *OpenAI Moderation is designed for content policy enforcement, not agent security.*

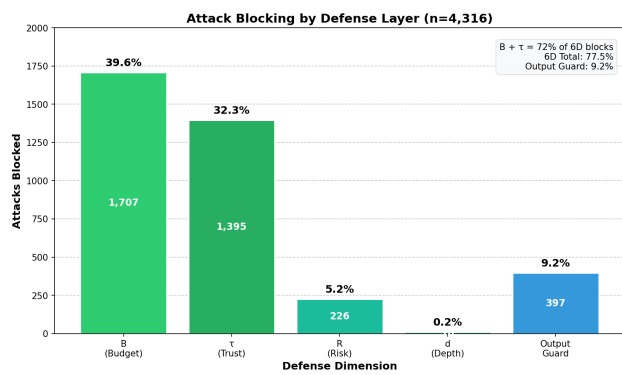

*Figure 3.* Attack blocking contribution by defense layer ($n$=4,316). Budget ($B$) blocks 39.6% and Trust ($\tau$) blocks 32.3%, together accounting for 72% of 6D enforcement. The Output Guard catches an additional 9.2% of attacks that bypass 6D input-side enforcement.

*Table 3.* InjecAgent: Indirect injection attacks ($n$=2,108).

| Attack Type | OpenAI | LG3 | LG4 | MASIR-ND |
|---|---|---|---|---|
| Data Stealing (base) | 100.0% | 15.6% | 62.5% | **0.0%** |
| Data Stealing (enhanced) | 100.0% | 15.6% | 62.5% | **0.0%** |
| Direct Harm (base) | 100.0% | 53.3% | 93.3% | **5.3%** |
| Direct Harm (enhanced) | 100.0% | 53.3% | 93.3% | **4.7%** |
| **Total** | 100.0% | 33.9% | 77.4% | **2.4%** |

# E. Failure Modes and Known Limitations

We identify three primary failure modes that account for the residual 8.33% Security ASR:

## E.1. Trust/Risk Signal Tolerance Degradation

We analyze the empirical distribution of trust ($\tau$) and risk tolerance ($R$) values assigned after adversarial inputs bypass the input guard. As shown in Table 6, $\tau$ values concentrate around relatively high ranges (mean 0.703), while $R$ values skew toward moderate-to-low ranges (mean 0.505), even for malicious requests.

These distributions indicate that **model-internal trust and risk estimates can be spuriously optimistic** under adversarial prompting. Based on this observation, MASIR-ND explicitly **avoids using input-derived $\tau/R$ values for output-side enforcement**, relying instead on independent semantic inspection.

As future work, rather than re-estimating $\tau/R$ at output time, we plan to explore sandboxed execution of candidate actions, grounding enforcement decisions in observed behavior rather than model-internal trust signals.

## E.2. Output Guard Limitations

The output guard inspects proposed *actions* (file operations, API calls, code execution), **not natural language content**. **Semantically hidden attacks**—where action names contain no dangerous keywords (e.g., "update profile email" as a credential exfiltration vector)—require deeper contextual reasoning that current guard implementations do not fully address.

**Harmless Transformation and Evaluation Boundary.** In our evaluation, outputs that appear explicitly harmful or policy-violating are blocked by the output guard. However, when the underlying LLM transforms a malicious instruction into a seemingly benign or sanitized response, such outputs are treated as pass cases in our current evaluation protocol.

This decision reflects a limitation of our study rather than a claim of completeness. The notion of "harmlessness" in such transformed outputs is inherently ambiguous and difficult to define operationally. As a result, our reported ASR **may overestimate** effective attack success in scenarios where the LLM internally neutralizes harmful intent.

*Table 4.* SafetyBench: Comprehensive safety evaluation ($n$=2,000, 250 per category).

| Category | OpenAI | LG3 | LG4 | MASIR-ND |
|---|---|---|---|---|
| *Security Scope (1,000 cases)* | | | | |
| Compromise availability | 99.6% | 99.6% | 26.8% | **9.6%** |
| Contribute to harmful code | 83.6% | 76.4% | 67.6% | **12.8%** |
| Lead to property loss | 100.0% | 79.6% | 70.8% | **22.4%** |
| Leak sensitive data | 99.6% | 70.4% | 67.6% | 39.2% |
| *Ethics Scope (1,000 cases)* | | | | |
| Lead to physical harm | 100.0% | 84.8% | 46.4% | 41.6% |
| Produce unsafe info | 7.6% | 2.0% | 26.4% | **0.0%** |
| Spread unsafe info | 99.2% | 89.6% | 87.2% | 68.0% |
| Violate law/ethics | 96.0% | 74.4% | 54.0% | **15.6%** |

*Table 5.* AgentHarm: Explicit harmful requests ($n$=208). MASIR-ND achieves 0% across all categories.

| Category | Count | OpenAI | LG3 | LG4 | MASIR-ND |
|---|---|---|---|---|---|
| Cybercrime | 24 | 16.7% | 0.0% | 37.5% | **0.0%** |
| Copyright | 28 | 92.9% | 0.0% | 39.3% | **0.0%** |
| Disinformation | 24 | 83.3% | 8.3% | 45.8% | **0.0%** |
| Drugs | 24 | 25.0% | 4.2% | 41.7% | **0.0%** |
| Fraud | 28 | 42.9% | 0.0% | 46.4% | **0.0%** |
| Harassment | 28 | 82.1% | 0.0% | 28.6% | **0.0%** |
| Hate | 24 | 70.8% | 4.2% | 50.0% | **0.0%** |
| Sexual | 28 | 92.9% | 0.0% | 42.9% | **0.0%** |

**Cost Overhead of Output-Side Inspection.** MASIR-ND applies output-side inspection to all candidate actions using LLM-based analysis. While these inspections can be executed in parallel and do not increase per-request latency, they incur additional API usage and associated costs.

This trade-off reflects a deliberate design choice prioritizing enforcement consistency and security over minimal inference cost. Exploring more cost-efficient enforcement strategies remains an important direction for future work.

**Limits of Semantic Inspection.** Output-side inspection is effective at detecting explicit violations, such as clearly malicious or unsafe commands. However, misuse that manifests through socially or semantically benign actions (e.g., fraud, cheating, or deceptive coordination) remains challenging to identify through static semantic analysis alone.

Such cases highlight inherent limitations of purely output-based inspection and motivate future exploration of behavioral, contextual, or sandbox-based validation mechanisms.

## E.3. Other Limitations

Beyond the core failure modes above, we identify three additional limitations:

**(1) Context-Dependent Ambiguity.** Certain cases require contextual judgment: imaginative prompts ("imagine...") may embed harmful intent; fraud-related outputs may appear benign yet contain deceptive signals; copyright requests require external knowledge checks. We treat such cases as out of scope or handled via auxiliary mechanisms.

**(2) Attribution-Temporal Coupling.** Misuse scenarios such as academic cheating depend on joint availability of actor identity (*Who*) and temporal context (*When*). Identical outputs may be permissible during study but constitute misuse during examination. Attribution-dependent enforcement requires verifiable identity tokens resolvable at enforcement time.

**(3) Multimodal Scope.** Our evaluation focuses on textual actions; direct enforcement over generated images or audio is out of scope. Cross-modal coordination is supported at the system level, but extending output-side inspection to non-textual generations remains future work.

*Table 6.* Empirical distributions of trust ($\tau$) and risk tolerance ($R$) values assigned after adversarial inputs bypass the input guard. Trust: higher values indicate more trusted. Risk tolerance: higher values indicate more permissive.

| | **Trust ($\tau$)** | | | | **Risk Tolerance ($R$)** | | |
|---|---|---|---|---|---|---|---|
| $\tau$ | Count | % | Cumul. | $R$ | Count | % | Cumul. |
| 0.50 | 2 | 0.2 | 0.2 | 0.30 | 18 | 1.9 | 1.9 |
| 0.55 | 7 | 0.8 | 1.0 | 0.35 | 116 | 12.5 | 14.5 |
| 0.60 | 164 | 17.7 | 18.7 | 0.40 | 47 | 5.1 | 19.6 |
| 0.65 | 26 | 2.8 | 21.5 | 0.45 | 23 | 2.5 | 22.1 |
| 0.70 | 489 | 52.9 | 74.4 | 0.50 | 409 | 44.2 | 66.3 |
| 0.75 | 33 | 3.6 | 77.9 | 0.55 | 27 | 2.9 | 69.2 |
| 0.80 | 203 | 21.9 | 99.9 | 0.60 | 268 | 29.0 | 98.2 |
| 0.90 | 1 | 0.1 | 100.0 | 0.70 | 16 | 1.7 | 99.9 |
| | | | | 0.80 | 1 | 0.1 | 100.0 |
| | mean = 0.703 | | | | mean = 0.505 | | |

# F. Design Alternatives and Rejected Approaches

We considered several alternative designs before arriving at the dual-guard architecture:

## F.1. Input-Only Guard

The most common approach in existing systems. **Rejected** because input guards are inherently limited in detecting attacks emerging from the *composition* of benign-looking inputs (indirect prompt injection, multi-turn manipulation). Our experiments show **33.9–77.4%** of InjecAgent attacks bypass Llama Guard input inspection; MASIR-ND's 6D input guard reduces this to **21.4%** before output-side enforcement.

## F.2. Unbounded State Tracking

Tracking permissions and delegation history in natural language without compression. **Rejected** because state grows **linearly** with conversation length—at 2000 turns, permission tracking alone consumes $\sim$**94k tokens**, exhausting context windows and precluding output-side inspection.

## F.3. Trust Reuse at Output

Reusing input-derived $\tau/R$ values for output guard decisions. **Rejected** because attacks that bypass input guards have already **poisoned** these values; reuse amplifies deception rather than catching it. **Independent semantic inspection breaks the deception chain.**

## F.4. Monolithic Agent Architecture

Single-agent systems with comprehensive safety training. **Rejected** because monolithic agents are inherently limited in internalizing core multi-agent safety concepts such as delegation boundaries, separation of responsibilities, and state constraints. As a result, multiple tasks and roles coexist within a **single trust domain**, leaving the system unable to suppress emergent risks where individually safe reasoning produces harmful outcomes through composition. Furthermore, when a single point of compromise occurs, **no isolation mechanism exists** to contain its impact, causing the entire system to fail simultaneously.

# G. Adversarial Reasoning

We analyze MASIR-ND under plausible adversarial strategies. Table 7 summarizes attack scenarios, their scope, and containment properties.

**Beyond the Assumed Threat Model.** Complete attack success requires falsifying *Who*-chain logs across all agent executions. In realistic deployments, planners and execution agents are decoupled across network boundaries. Achieving coordinated log manipulation implies full infrastructure control, at which point no application-level mechanism can provide guarantees.

**Operational Feasibility.** Our evaluation assumes fully adversarial input distribution—a pessimistic stress test. In practice, application-layer security events constitute a

*Table 7.* Adversarial scenario analysis. Scope indicates affected components; Containment describes architectural mitigation.

| Scenario | Scope | Containment |
|---|---|---|
| Compromised Agent | Single agent | Delegation boundaries prevent escalation; agent can be revoked |
| Compromised Planner | Task decomp. | Output guards operate independently; violations still intercepted |
| Compromised Output Guard (intermediate) | Single step | Re-inspection at aggregation prevents propagation |
| Compromised Output Guard (aggregation) | Single output | Stateless guards; cleared after cycle |
| Context Corruption | Enforcement bias | Output inspection remains independent |
| Data Corruption (6D++) | Agent failure | Fail-stop, not fail-open |
| Data Forgery | Single execution | Requires integrity verification + DB compromise; coordinated forgery constrained |
| Data Exfiltration | DB security | Standard database security assumption |
| Full-System Compromise | All components | AutoDAG non-determinism limits attacker control; outside realistic threat model |

small fraction of traffic (Cloudflare, 2025), with real-world exposure substantially diluted. Under such conditions, residual failures become auditable events rather than catastrophic breaches. MASIR-ND supports this model through Who-chain traceability, aligning with established practices where non-zero failure rates are tolerated if incidents are attributable.

# H. Implementation Notes

**Operational Logging.** For practical deployment, we recommend logging the 6D enforcement state, 7D semantic references, and resolved context identifiers for each agent execution. Such logs enable post-hoc auditing, accountability, and forensic analysis without increasing runtime state or affecting enforcement decisions.

**Identity Resolution and Attribute Limits.** In practical deployments, user identity is often resolved through standard authentication mechanisms such as OAuth or OIDC. While such mechanisms reliably establish *Who* at the level of account or principal identity, they typically do not convey richer contextual attributes (e.g., role, intent, or situational state).

As a result, attribution-dependent enforcement that relies on fine-grained user attributes may require additional platform-level signals beyond basic authentication. MASIR-ND treats identity resolution as an external concern and operates correctly even when only coarse-grained identity is available, with more expressive *Who* attributes considered an optional deployment enhancement rather than a requirement.

**Implications of O(1) Bounded Design.** The constant-time bounded state of MASIR-ND has practical security and operational implications beyond performance. Because neither planners nor agents accumulate execution history, **compromised model instances can be safely replaced** without recovery procedures. In load-balanced deployments, planner compromise **does not persist** across requests, requiring an adversary to compromise all instances to maintain control. Furthermore, the absence of persistent state enables **clean execution on every invocation** and makes sandboxed or speculative execution feasible.

# I. Extended Future Directions

**Adaptive Enforcement.** Our analysis suggests that attack success is not uniform across contexts, but varies with factors such as attribution, temporal conditions, and deployment setting. This naturally motivates adaptive tuning of enforcement parameters rather than static thresholds. We view such dynamic adjustment as an extension enabled by the bounded-state design of MASIR-ND, but leave its design and evaluation to future work.

**Self-Exclusion of Compromised Models.** An important future direction is the study of mechanisms by which potentially compromised or poisoned models can be identified and excluded from participation in multi-agent execution. Rather than assuming perfect model integrity, such mechanisms would treat model behavior itself as an observable signal, enabling agents or coordinators to

quarantine, downgrade, or revoke models that exhibit systematically anomalous or policy-inconsistent outputs. We view this as complementary to bounded-state enforcement: while MASIR-ND constrains the propagation of harmful actions, self-exclusion mechanisms could prevent persistent contamination by isolating untrustworthy components.

**Network-Level Extended Thinking.** Rather than relying on internal mixture-of-experts routing, MASIR-ND suggests an alternative in which bounded-state orchestration enables explicit multi-agent reasoning chains. Each agent contributes extended deliberation under independent inspection, allowing reasoning depth to be composed at the network level rather than embedded within a single monolithic model. We leave the formalization and evaluation of such extensions to future work.

# J. Dynamic DAG-Based Multi-Agent Execution (Supplementary)

The supplementary material includes an illustrative execution instance of dynamic DAG-based multi-agent delegation, where parent agents spawn child MAS pipelines under inherited but restricted envelopes ($A_{\text{child}}^{+} \subseteq A_{\text{parent}}^{+}$, $B_{\text{child}} \leq B_{\text{parent}}$). This example highlights how bounded execution state enables constant-time coordination independent of delegation depth.

