# OpenReview forum: "Position: Bounded Execution State Enables Practical Security for Multi-Agent LLM Systems"
_ICML.cc/2026/Position_Paper_Track — Submitted to ICML 2026 Position Paper Track_

### Official Review · Reviewer_Hn4Y · 2026-03-13

**Significance:** 3
**Argument Clarity:** 3
**Rating:** 3
**Confidence:** 2

**Questions:**

The authors present that output side verification to detect composition attacks hasn't been implemented in multi-agent systems due to tractability. However, I would have appreciated if the authors provided additional context on - how does it compare with single agent systems? Are there anything with respect to multi-agent systems that make it unique? Beyond the simple change in threat model of "inject adversarial content into inter-agent communications" which anyway seems orthogonal to the solution presented?

**Alternative Views Section:**

Yes

**Compliance With Llm Reviewing Policy A Conservative:**

Affirmed.

**Discussion Potential:**

1

**Paper Summary:**

The paper presents that bounded execution state that allows for O(1) inspection independent of sequence length, and explicit output side enforcement are both critical for safe multi agent systems. A bounded 13-dimensional (13D) state with six enforcement and seven semantic dimensions for constant-time inspection with a constant-size state is presented. These 13 dimensions can be maintained in a 64-byte envelope regardless of conversation length or delegation depth.

**Position:**

Yes

**Position In Title:**

Yes

**Related Work:**

3

**Strengths And Weaknesses:**

Thank you for submitting. Although I appreciate reading the paper, my critical concern is - if this should be a position paper or a paper in the conference? To be more concrete, in the introduction the authors clearly clarify that "Although dual-boundary enforcement may appear intuitive in hindsight, output-side guards were deliberately avoided to prevent state explosion in prior MAS architectures. Our
contribution is identification of the bounded state, a missing prerequisite that enables a practical dual-guard architecture". If this is true, than although this is presented as a position paper, I am inclined to suggest that this isn't really presenting a position that would lead to greater discussion in the community, but is rather a very clever solution to the proposed problem instead.

Strengths:
+ A very relevant problem domain with solution that is complete.
+ O(1) inspection that is independent of sequence length.
+ Clearly defined thread model (Section 2.2)

Weaknesses:
- I am NOT fully sure, what the formal guarantees of the output guard are? The authors mention "Instead, the output guard performs independent semantic inspection, breaking the deception chain." although this is true, then the assumptions need to be updated that the output guard can potentially stochastic? The paper's Coq verification (29 theorems) proves structural properties of the bounded state-envelope size, scalability, non-escalation of permissions-but not the classification accuracy of the output guard itself.
- From the context of a position paper, this paper is NOT accessible to the ICML readers given the strong assumptions made in the introduction. I would encourage the authors to carry along the researchers who might be tangentially relevant to the field. Please define your abbreviations the first time you use them. E.g. ASR in line 70.
- The authors mention (Section 10) that automated A- generation via substring matching causes up to 21% degradation in UAR. Combined with not relying on tau/r - since it's suceptible to prompt-injection attacks makes me wonder, how practical is the solution?

**Support:**

3

---

### Official Review · Reviewer_5A5B · 2026-03-13

**Significance:** 4
**Argument Clarity:** 3
**Rating:** 5
**Confidence:** 4

**Questions:**

1.Regarding the O(1) execution claim in Section 3.2: While the bitmap operations are indeed O(1), how do you plan to handle the database lookup latency for resolving ref_7D and ref_ctx? Do you have any micro-benchmarks showing how this affects system throughput when scaling to thousands of concurrent agents?
2.In Section 4.2, how resilient is the initial 6D policy generation phase? If an attacker manages to trick the Input Guard's LLM into generating overly permissive bounds (e.g., assigning a high trust score τ to a malicious payload), wouldn't the output guard simply execute the actions based on that flawed policy?

**Alternative Views Section:**

Yes

**Compliance With Llm Reviewing Policy A Conservative:**

Affirmed.

**Discussion Potential:**

4

**Paper Summary:**

The authors present a position paper arguing that securing Multi-Agent Systems (MAS) requires a fundamental shift from input-only filtering to explicit output-side enforcement. To make output inspection computationally feasible at scale, they propose MASIR-ND, which compresses the execution state into a bounded 64-byte, 13-dimensional envelope. The paper supports its position through extensive empirical evaluation (showing a reduction in security ASR to 8.33% compared to Llama Guard 3's 48.82%) and backs its scalability claims with 29 mechanized proofs in Coq.

**Position:**

Yes

**Position In Title:**

Yes

**Related Work:**

3

**Strengths And Weaknesses:**

Strengths:
•Formal Verification: I really appreciated the inclusion of Coq proofs (Section 5.1). It is quite rare to see formal verification applied to LLM agent architectures in this venue. This provides a rigorous theoretical foundation for their claims about unbounded delegation depth (Theorem 5.3).
•Compelling Argument against Input-Only Guards: The critique of existing input filters is very well articulated. Section 4.1 (specifically the comparison between LG3 and LG4 on different attack types) effectively demonstrates the inherent trade-offs in current classifier designs and builds a strong case for why output-side enforcement is needed for composition attacks.
•Solid Evaluation: The scale of the empirical validation (over 90,000 inference evaluations) is impressive for a position paper. Table 1 clearly separates security ASR from content moderation, which is a necessary distinction for MAS deployments.
Weaknesses:
•Practical I/O Bottlenecks: The authors heavily emphasize the O(1) nature of the 64-byte inline check (Equation 1 in Section 3.2). However, this theoretical O(1) somewhat masks the practical system-level overhead. Since the architecture offloads the 7D semantics and context to an external database via UUIDs, querying this database during the execution phase will inevitably introduce I/O latency. In a real-world, highly parallelized MAS, this database lookup could become a severe bottleneck.
•Vulnerability in Policy Bootstrapping: According to Section 4.2, the first phase of the Input Guard uses an LLM to derive the 6D policy boundaries (including trust τ and risk R). If an attacker successfully uses prompt injection during this very first generation phase, the foundational bounds of the envelope could be poisoned from step one.
•Contextualization in Systems Security: While the paper mentions Anderson (1972) in Section 7, the proposed "envelope" mechanism shares deep conceptual similarities with object-capability models in classical operating systems. A brief discussion bridging this LLM-specific work with traditional capability-based security paradigms would strengthen the paper's theoretical positioning.

**Support:**

4

---

### Official Review · Reviewer_EbSV · 2026-03-15

**Significance:** 1
**Argument Clarity:** 1
**Rating:** 2
**Confidence:** 4

**Questions:**

* Why have permitted and forbidden action categories? Is it possible that these would be difficult to specify and context specific?
* How does MASIR-ND handle decomposition attacks?
* How does the constant-sized envelop handle unbounded chains of tool calls?
* Are any of the results on multi-agent systems? What's the unique difference when trying to secure multi-agent systems?

**Alternative Views Section:**

Yes

**Compliance With Llm Reviewing Policy A Conservative:**

Affirmed.

**Discussion Potential:**

2

**Paper Summary:**

This paper considers the problem of securing agentic systems from malicious adversaries. The paper identifies two flaws with input-only safeguards. The paper proposes a framework for constant-sized flagging and both input and output classifiers. The paper provides a detailed explanation of a proposed approach.

**Position:**

Yes

**Position In Title:**

Yes

**Related Work:**

2

**Strengths And Weaknesses:**

#### Strengths
* The paper is considering an important problem: safeguarding multi-agent or agentic systems.

#### Weaknesses
* The paper jumps right into specifics of some architecture and design choices without a lot of context. There is a lot of implementation and design choice details without any evidence for the position (or against what the status quo for monitoring agents).
* This paper does not consider a sufficiently general array of possible solutions. The authors propose one very specific solution and claim by fiat that this solves problems that are proposed by the authors.
* On lines 204-205, the authors claim that "Input guard filtering is fast but incomplete and prone to false negatives", but there is no evidence provided for this. There are other claims throughout the paper that are not supported by evidence (i.e. citations, results, etc.).
* Section 4.1 does not answer the question why do single-layer checks fail?
* There is no support for why a constant sized envelope is necessary or sufficient.
* Section 5 has a bunch of theorems without definitions or proofs...
* The claim that constant time inspection is possible is unsubstantiated.
* The proposed approach and experiments are not on multi-agent systems, but the entire motivation for the paper is multi-agent systems (at least what is discussed in the abstract and the intro).

#### Minor
* A lot of abbreviations are used that are not defined throughout, making the paper more difficult to read (e.g. "NL")

**Support:**

1

---

### Decision · Program_Chairs · 2026-04-30

**Decision:**

Reject

**Comment:**

The paper tackles an important problem, and while one reviewer was strongly positive  about both the architectural insight and the supporting empirical and formal evidence, other reviewers raised substantial concerns about accessibility, generality, positioning as a position paper, and whether the central claims are argued broadly enough beyond one specific framework. For this track, the key question is not only whether the idea is interesting, but whether the paper makes a broadly compelling case for what the community should adopt, and I do not think the current version fully clears that bar. While I see promise in the core viewpoint, the overall case is not strong enough for acceptance this round.